# HIERARCHICAL W-LEARNING

## ABSTRACT

Inspired by a model of the brain called projective simulation, which has attracted interest among physicists in recent years, we develop a simple and generic new method for hierarchical reinforcement learning in this article. The proposed method generalizes the action-value Q function to W function, enabling the agent to execute actions according to a hierarchical strategy. In the first part of the article, we present a rigorous construction of the hierarchical structure, along with the W-learning algorithm and the hierarchical policy gradient theorem. In the second part, as an example, we illustrate the W-learning procedure in the context of a navigation task. Experimental results show that the introduction of the hierarchical structure can lead to better performance than traditional Q-learning, provided the strategy is well designed and the update parameters are appropriately chosen. Various policy gradient methods are also investigated.

## 1 INTRODUCTION

The human brain is one of the most complicated and fascinating objects in nature, and many people believe that building a proper model of it may finally lead to artificial intelligence. In recent years, inspired by the study of path integrals in quantum mechanics, physicists have constructed a model for the brain called projective simulation (Briegel, 2012; Briegel & De las Cuevas, 2012; Paparo et al., 2014; Melnikov et al., 2017; Dunjko & Briegel, 2018; Boyajian et al., 2020; Flamini et al., 2024) which aims to keep track of the agent's memory and make the agent learn from the experience. The basic unit in the projective simulation is the clip units and the transition between the clips are described by the transition probability shown in Fig. 1. As we can see, the clip structure is a model

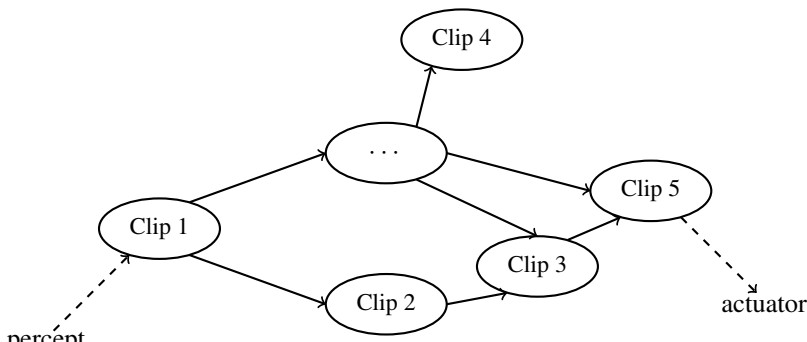

Figure 1: Clip structure of the brain in projective simulation

for the neural network in the brain. The clip could represent any information stored in the brain such as an action, a state, or a series of actions and states. The learning procedure starts from a signal that simulates a percept, and then the signal will go through the entire clip structure following the probability distribution described by the arrows in the figure. Finally, the signal will reach a clip that simulates an actuator, leading to the action of the agent in the real physical world. After each percept-actuator cycle, the structure of the clips along with the transition probability, will update according to the feedback given by the environment where the agent is embedded.

There are various conceptual discussions about the creation and annihilation of the clips, as well as the dynamic evolution of the clip structure. Here, in this article, we will study the update of the

transition probability in the context of reinforcement learning (Sutton & Barto, 1998) and focus on the hierarchical structure of those clips.

The traditional reinforcement learning describes the behavior of the agent as a probability distribution over the actions based on the state, and the goal of the training procedure is to determine the optimal policy which maximizes the expected reward following it. Here the clip structure focuses more on the internal structure of the agent, and at the same time, the evolution of such structure is determined by the accumulated reward. There is already a significant amount of work on the study of the hierarchical reinforcement learning (Sutton et al., 1999b; Dieterich, 2000; Hengst et al., 2002; Botvinick, 2008; Bacon et al., 2017; Zhou et al., 2019; Hutsebaut-Buysse et al., 2022) and here we focus on the hierarchical generalization of $Q$-learning (Watkins et al., 1989; Watkins & Dayan, 1992). In our work, the action-value function $Q$ is extended to the strategy-value called the $W$ function in order to reflect the hierarchical structure of the strategies, also we can see that there is no need to construct a termination function and the update rule is a simple repetition of the update rule for $Q$-learning at each layer. Moreover, in the $W$-learning, the agent could be taught to learn both the policy-over-option and the intra-option, therefore the method is more flexible and could be applied to deal with more dynamical tasks. In fact, it is possible to learn all the tasks provided that the strategy could be represented in terms of a hierarchical clip structure. Finally, we consider the experimental implementation of such an algorithm in a navigation task and compare the results with $Q$-learning. From the experiment, we see that $W$-learning performs much better than the $Q$-learning if the strategy is properly designed and it is also interesting to note that the assignment of the higher level update parameters will also play an important role in practice.

## 2 HIERARCHICAL STRUCTURES

In the context of reinforcement learning, we write the agent state as $s_t$ and the agent chooses to perform the action $a_t$ based on the policy $\pi(\cdot \mid s_t)$ at each step $t$. After executing the action $a_t$, the agent then moves on to the step $t + 1$ and transforms to the next state $s_{t+1}$ obtaining a reward denoted as $r_{t+1}$. During the whole task, to evaluate the performance of the agent given the strategy $\pi$, one can define the state-value function $V_\pi(s)$

$$V_\pi(s) = \mathbb{E}_\pi \left\{ \sum_{t=0}^\infty \gamma^t r_{t+1} \mid s_0 = s \right\}, \tag{1}$$

where $\gamma \in [0, 1)$ is the discount factor of the rewards. From the above definition, one can see that the value function tells us the expectation of the total reward during the whole process following the policy $\pi$ if the initial state is set as $s$. Moreover, one can also define the action-value function as

$$Q(s, a) = \mathbb{E}_\pi \left\{ \sum_{t=0}^\infty \gamma^t r_{t+1} \mid s_0 = s, a_0 = a \right\} \tag{2}$$

which gives us the expectation of the accumulated reward if the initial state and action are set as $s_0 = s, a_0 = a$.

We now study the hierarchical structure of the policies, where the strategy at the $l^{th}$-level is denoted as $\omega_l$. The policy $\pi$ can be expressed as

$$\pi_\theta(a \mid s) = \sum_\omega \pi_{\theta_{l-1}}(a \mid \omega_{l-1}) \cdots \pi_{\theta_i}(\omega_{i+1} \mid \omega_i) \cdots \pi_{\theta_0}(\omega_1 \mid s) \tag{3}$$

where $\pi_{\theta_i}(\omega_{i+1} \mid \omega_i)$ tells us the probability of choosing the strategy $\omega_{i+1}$ at the level $i + 1$ given that the strategy of the $i$ th level is chosen as $\omega_i$. The parameters $\theta_i$ are introduced here to describe the behavior of the strategies for some fixed basis[1]. We should note that the strategy $\omega_i$ at the level $i$ and the parameter $\theta_i$ are in fact sets of vectors $\theta_i = (\boldsymbol{\theta}_i^1, \cdots \boldsymbol{\theta}_i^k)$ and $\omega_i = (\omega_i^1, \cdots, \omega_i^k)$ where $k$ is the number of possible strategies at the level $i$. Based on the definition of the strategy and the corresponding hierarchical structure, we can write down the strategy-value function as

$$W(s, \omega, a) := \mathbb{E}_\pi \left\{ \sum_{t=0}^\infty \gamma^t r_{t+1} \mid s_0 = s, \omega_0 = \omega, a_0 = a \right\}, \tag{4}$$

---

[1]More precisely, the policy $\pi_\theta(a \mid s)$ is given by the soft-max distribution $\pi_\theta(a \mid s) = \frac{\exp \boldsymbol{\theta}^t \cdot \mathbf{x}(a,s)}{\sum_b \exp \boldsymbol{\theta}^t \cdot \mathbf{x}(b,s)}$, where $\mathbf{x}(b, s)$ are the corresponding basis expressed as a vector.

which gives us the expectation of the total reward if the initial state, strategy and action are chosen as $s, \omega, a$. Moreover, given the strategy-value function, one can also define the $W$ value at the level $l - 1$ as

$$W(s, \omega_1, \ldots, \omega_{l-1}) = \sum_a \pi_{\theta_{l-1}}(a \mid \omega_{l-1})W(s, \omega_1, \ldots, \omega_{l-1}, a) \tag{5}$$

which tells the expectation of the total reward if the initial state and strategy is set as $s$ and $\omega = (\omega_1, \ldots, \omega_{l-1})$. The hierarchical $W$ value between the $l - 1$ and $l - 2$ has the recursion relation

$$W(s, \omega_1, \ldots, \omega_{l-2}) = \sum_{\omega_{l-1}} \pi_{\theta_{l-2}}(\omega_{l-1} \mid \omega_{l-2})W(s, \omega_1, \ldots, \omega_{l-2}, \omega_{l-1}). \tag{6}$$

We can also define the weight function $\mu_{\omega,\theta}(s) = \pi_{\theta_{l-2}}(\omega_{l-1} \mid \omega_{l-2}) \cdots \pi_{\theta_0}(\omega_1 \mid s)$, which gives the probability that the agent chooses the strategy $\omega$ at the state $s$. Moreover, according to the definition of $V(s)$ and $W(s, \omega, a)$, we have the relation

$$W(s, \omega, a) = r(s, \omega, a) + \gamma \sum_{s'} p(s' \mid s, \omega, a)V(s') \tag{7}$$

where $r(s, \omega, a) = \mathbb{E}\{r_{t+1} \mid s_t = s, \omega_t = \omega, a_t = a\}$ and $p(s' \mid s, \omega, a)$ is the probability of $s_{t+1} = s'$ when the state, strategy and the action of the $t$ step is chosen as $(s, \omega, a)$. By summing over all the strategies $\omega$, weighted by $\mu$, we will obtain the relation between $V$ and $Q$, familiar from the relation introduced in the standard reinforcement learning literature.

The goal of the learning procedure is to determine the optimal policy $\pi^*$ that has the maximum total value function $V^*(s)$ which satisfies the Bellman optimality equation. And the stochastic approximation of Bellman operator is to follow the update rule

$$Q(s_t, a_t) \longleftarrow Q(s_t, a_t) + \alpha(t)(r_{t+1}(s_t, a_t) + \gamma \max_a Q(s_{t+1}, a) - Q(s_t, a_t)) \tag{8}$$

where the parameter $\alpha(t)$ is the learning rate and $\gamma$ is called the discount factor. It has been proven that the $Q$ value will converge to the maximum $Q^*$. For the hierarchical structure of the policy, based on the formula equation 8, here we introduce the update rule for the $W$ function written as

$$W_k(s_t, \omega_t) \longleftarrow W_k(s_t, \omega_t) + \alpha_k(t)(g_k(s_t, \omega_t, a_t, r_{t+1}) + \gamma_k \max_\omega W_k(s_{t+1}, \omega) - W_k(s_t, \omega_t)) \tag{9}$$

where the parameters $\alpha_k(t)$ are learning rate for the $k$ layer in the strategy hierarchical structure and, for simplicity, $W(s, \omega_1, \ldots, \omega_k)$ is denoted as $W_k(s, \omega)$. $g_k$ is the parameter that determines the update for the higher level $W$ function and it depends on the state, action $a_t$, strategy $\omega_t$ and the reward $r_{t+1}$. Later in the experiment, we will see that it plays a crucial role for the performance of the agent. Although in practice it is more convenient to determine the value $W$, we should note that there is no one-to-one correspondence between $W$ and the policy $\pi$ together with the strategies. Given the learned $W$ function, one can determine the strategy layer by layer following the rule

$$\omega_i^* = \arg\max_{\omega_i} W(s, \omega_1^*, \ldots, \omega_{i-1}^*, \omega_i) \tag{10}$$

therefore the policy is determined as $(s, \omega_1^*, \ldots, \omega_{l-1}^*, a^*)$, and the algorithm can be described as below.

---

**Algorithm 1**: Hierarchical $W$ Learning

---

Algorithm parameters: learning rate and discount factors $\alpha_i, \gamma_i \in (0, 1]$
Initialize $W(s, \omega, a)$ for all $s$, $\omega$ and $a$

Loop for each episode:
    Initialize $s_0$ and $t = 0$
    Loop for each step of episode:
        Choose the strategy $\omega_t$ from $s_t$ using the policy derived from $W$
        Then determine and execute the action $a_t$, observe $r_{t+1}, s_{t+1}$
        Choose the strategy $\omega_{t+1}$ using the $W$ function
        For all $k$: $W_k(s_t, \omega_t) \longleftarrow W_k(s_t, \omega_t) + \alpha_k(g_k + \gamma_k \max_\omega W_k(s_{t+1}, \omega) - W_k(s_t, \omega_t))$
        $t = t + 1$
    until $s_t$ is terminal

---

As we can see, in the context of reinforcement learning, the key is to figure out an optimal policy as a probability distribution $\pi^*(s, \omega, a)$, which gives the maximum expected reward $V^*(s)$ throughout the training process starting from the state $s$. However, although the method for updating the function $W$ provides us a convenient way to obtain an optimal policy, it is still necessary to study and keep track of the update of the policy in a straightforward way. In our setup, given the basis, the policy update now becomes the update for the parameters $\theta$, written as

$$\theta_i^{(t+1)} = \theta_i^{(t)} + \alpha_i(t) \frac{\partial V(s)}{\partial \theta_i} \tag{11}$$

where $\partial_\theta V$ is the gradient for the value function giving the direction for the $\theta$ leading to a greater value and $\alpha_i(t)$ is the learning rate for the policy. The superscripts $t$ are used to label the training step.

Now, we present the evaluation of the gradient $\partial_\theta V(s)$ in a precise way based on the generalization of policy gradient method for the policy without hierarchical structures (Sutton et al., 1999a). From the recursion relation equation 6, we have the derivative for the value function

$$\frac{\partial V(s)}{\partial \theta_i} = \frac{\partial}{\partial \theta_i} \sum_{\omega_1} W(s, \omega_1) \pi_{\theta_1}(\omega_1 \mid s) \tag{12}$$

$$= \sum_{\omega_1} \left( W(s, \omega_1) \frac{\pi_{\theta_1}(\omega_1 \mid s)}{\partial \theta_i} + \pi_{\theta_1}(\omega_1 \mid s) \sum_{\omega_2} \left( \pi_{\theta_2}(\omega_2 \mid \omega_1) \frac{W(s, \omega_1, \omega_2)}{\partial \theta_i} \right.\right.$$

$$\left.\left. + W(s, \omega_1, \omega_2) \frac{\partial \pi_{\theta_2}(\omega_2 \mid \omega_1)}{\partial \theta_i} \right) \right), \tag{13}$$

where we have used the relation two times, one from $V(s)$ to $W(s, \omega_1)$ the other from $W(s, \omega_1)$ to $W(s, \omega_1, \omega_2)$. Following such a procedure, one can write $W(s, \omega_1, \omega_2)$ in terms of $W(s, \omega_1, \omega_2, \omega_3)$ and then finally reach $V(s')$. At the same time, taking the relation

$$\frac{\partial \pi_{\theta_i}(\omega_{i+1} \mid \omega_i)}{\partial \theta_j} = 0 \tag{14}$$

for $j \neq i$ into consideration, we obtain

$$\frac{\partial V(s)}{\partial \theta_i} = \sum_{\omega_1, \ldots, \omega_i} \frac{\partial \pi_{\theta_i}(\omega_1, \ldots \omega_i \mid s)}{\partial \theta_i} W(s, \omega_1, \ldots, \omega_i) \tag{15}$$

$$+ \sum_{a, s'\omega} \pi_\theta(\omega, a \mid s) p(s' \mid s, \omega, a) \frac{\partial V(s')}{\partial \theta_i}.$$

where $\pi_{\theta_i}(\omega_1, \ldots, \omega_i \mid s) = \pi(\omega_i \mid \omega_{i-1}) \ldots \pi(\omega_1 \mid s)$. Summing over all the components, then we have

**Theorem 1 (Hierarchical Policy Gradient)** *In the context of hierarchical reinforcement learning described by the conditional probability $\pi(\omega_{i+1} \mid \omega_i)$, given all the strategy-value functions $W(s, \omega_1, \cdots \omega_i)$, the derivative of the value function $V(s)$ takes the form of*

$$\frac{\partial V(s)}{\partial \theta_i} = \sum_{s', \omega_1, \ldots, \omega_i} \sum_{k=0}^{\infty} \gamma^k \mathrm{Pr}(s \to s', k, \pi) W(s', \omega_1, \ldots, \omega_i) \frac{\partial \pi_{\theta_i}(\omega_1, \ldots, \omega_i \mid s')}{\partial \theta_i} \tag{16}$$

*where $\mathrm{Pr}(s \to s', k, \pi)$ is the probability that the agent is at the state $s'$ after $k$ steps.*

Based on the hierarchical policy gradient theorem, now we have the update method for the policy

$$\theta_i^{(t+1)} = \theta_i^{(t)} + \alpha_i(t) \sum_s d^\pi(s) \sum_{\omega_1 \cdots \omega_i} W(s, \omega_1, \ldots, \omega_i) \frac{\partial \pi_{\theta_i}(\omega_1, \ldots, \omega_i \mid s)}{\partial \theta_i} \tag{17}$$

where $d^\pi(s) = \sum_{t=0}^{\infty} \gamma^t \mathrm{Pr}(s_t = s \mid s_0, \pi)$. Moreover, we have

$$\theta_i^{(t+1)} = \theta_i^{(t)} + \alpha_i(t) \sum_s d^\pi(s) \sum_{\omega_1 \cdots \omega_i} f_u(s, \omega_1, \ldots, \omega_i) \frac{\partial \pi_{\theta_i}(\omega_1, \ldots, \omega_i \mid s)}{\partial \theta_i} \tag{18}$$

where $f_u$ satisfies the condition

$$\sum_s d^\pi(s) \sum_{\omega_1,\cdots,\omega_i} \pi(\omega_1,\ldots,\omega_i \mid s)(W(s,\omega_1,\ldots,\omega_k) - f_u(s,\omega_1,\ldots,\omega_k))\frac{\partial f_u}{\partial u} = 0 \qquad (19)$$

and

$$\frac{f_u(\omega_1,\ldots,\omega_i,a)}{\partial u} = \frac{\pi(\omega_1,\ldots,\omega_i, \mid s)}{\partial \theta_i}\frac{1}{\pi_{\theta_i}(\omega_1,\ldots,\omega_i \mid s)}. \qquad (20)$$

In fact, by substituting the policy with the given basis into equation 20, one can check that the function $f$ satisfies the relation $\sum_{\omega_i} \pi(\omega_i \mid \omega_{i-1})f(\omega_1,\ldots,\omega_i,a) = 0$, therefore, it is more reasonable to interpret the function $f$ as the hierarchical advantage function defined as $A(s,\omega_1,\ldots,\omega_{i+1}) = W(s,\omega_1,\ldots,\omega_i) - W(s,\omega_1,\ldots,\omega_{i-1})$.

In practice, during the training, one can use the sample at $S_t$, and $\Omega_t^i$ to approximate the sum over $s$ and $\omega$ in equation A thus we have

$$\theta_i^{(t+1)} = \theta_i^{(t)} + \alpha_i(t)W(S_t,\Omega_t^i)\frac{\partial \ln \pi_{\theta_i}(\Omega_t^i \mid S_t)}{\partial \theta_i} \qquad (21)$$

where $S_t$ and $\Omega_t^i$ are the value of the action the strategy at the step $t$. Moreover, one can also introduce the baseline $b_i(s)$ at each level to establish the hierarchical REINFORCE, REINFORCE with Baseline and Actor-Critic learning algorithms. Such generalization is straightforward and we present the algorithms in appendix A. The convergence for the update procedure equation 9 and equation 18 is shown in the appendix B.

## 3 EXPERIMENT

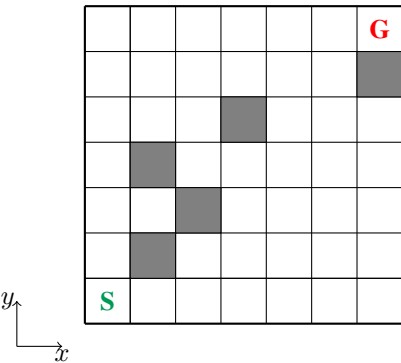

Figure 2: A 7-by-7 grid world with obstacles. The starting point and goal are denoted as S and G, respectively.

Now, as an example, we consider the navigation task to illustrate how the hierarchical $W$ learning works. In our case, the agent moves in a 7-by-7 grid world with obstacles, depicted as gray boxes in Fig.2. It starts from the bottom left corner and the goal is set at the top right corner, denoted as the green $S$ and red $G$, respectively. Moreover, during the training, the agent will get the reward of -10 if it hits the boundary or obstacles and will receive +20 if it reaches the goal. The agent receives -2 points for moving into an empty box. For convenience, we label the position of the box as $(x,y)$ following the convention of Cartesian coordinates and set the starting point $S$ as $(1,1)$.

In the context of reinforcement learning, now the state $s$ is described by a pair of integers $(x,y)$ and the action $a$ becomes the movement following the direction: right, left, up and down. In order to establish the hierarchical structure, here we introduce two strategies $\omega_1 = (\omega_1^1,\omega_1^2) = (\text{I}, \text{II})$ for the agent: strategy I is to move towards the goal while strategy II is to avoid the obstacles. Moreover, based on the observation that the starting point and the goal are located at two diagonal corners of the grid world, strategy I is then designed in the way that it will only lead to the right and up actions

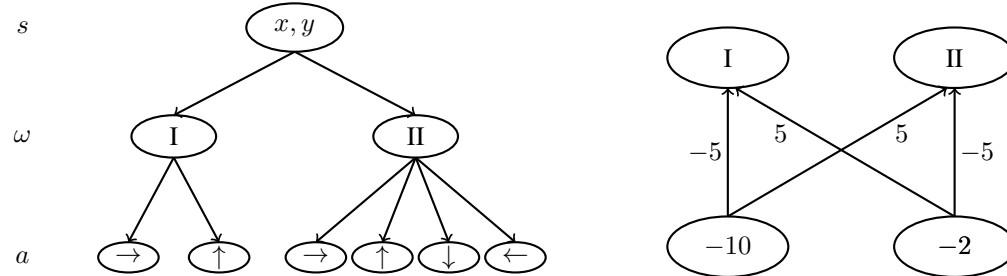

Figure 3: The hierarchical structure of the strategies is presented in terms of the clips.

while the strategy II will lead to all four actions in order to enable the agent to get rid of various kinds of obstacles.

Before conducting the experiment, we first introduce the rules for drawing a network for the clip structure. The network starts with the clip for the states $s$ and ends at the clips which represent the action $a$ of the agent. There are intermediate layers consisting of clips that represent the strategy $\omega$. Moreover, there could be arrows between two levels that describe the agent's decision procedure while no arrows are allowed between two clips inside a single layer. Following such rules, the clip structure for our navigation task can be depicted as the network shown in Fig.3. Here we should note that the design of the strategy is fixed before the training while the goal of training is to teach the agent what strategy to choose at each state. And, based on the real situation, the agent should be able to learn the action to take after fixing the strategy.

For the update rule, similar to the $Q$-learning, we set the low level update parameters equal to the rewards, i.e. $g_2 = r_{t+1}$. For the higher level update, we set the parameter as

$$g_1(\mathrm{I}, r = -2) = g_1(\mathrm{II}, r = -10) = 5 \qquad g_1(\mathrm{I}, r = -10) = g_1(\mathrm{II}, r = -2) = -5 \qquad (22)$$

based on the feedback machinery illustrated on the right side of Fig.3 where the absolute value is set as $|g_2| = 5$. That is, when the agent hits an obstacle and receives a reward of $-10$, strategy I receives feedback of $-5$, and strategy II receives feedback of $+5$. Conversely, if the agent goes into an empty box and receives the reward of $-2$, we assign a $+5$ feedback to the strategy I and $-5$ to the strategy II, which implies that the agent would be more likely to choose the strategy I when the next state is empty therefore, it will be guided to the goal in a faster way.

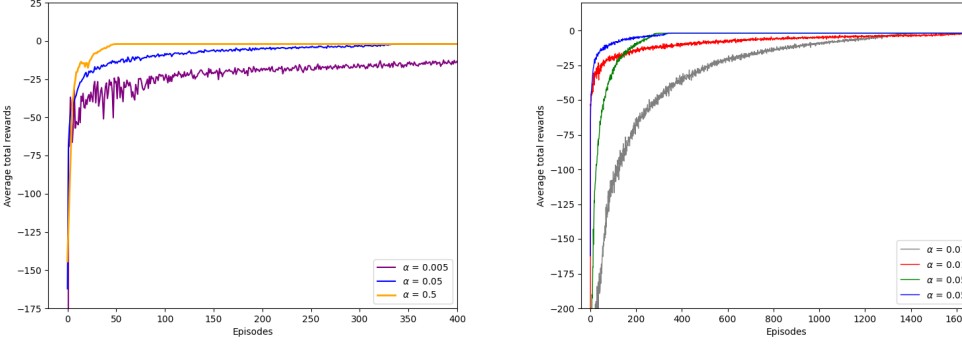

Figure 4: The $W$-learning results for different learning rates are shown on the left hand side of the figure and both of $Q$ and $W$ learning results are shown on the right. The reward is plotted by taking the average of 200 runs.

During the training, the discount factor is set as $\gamma = 0.99$ and we plot the $W$-learning results for $\alpha = 0.5$, $\alpha = 0.05$ and $\alpha = 0.005$ on the left side of Fig.4. From the figure, we can see the reward

curves approach the optimal level in a faster way for bigger learning rate and the critical point when the curve reach the optimal level is inversely proportional to the value of $\alpha$, from around 40 to 400 then to 4000. To investigate the effects of the hierarchical structure in a sensible way, we also plot the behavior of the reward for the $Q$-learning without the strategies and put them together on the right-hand side of Fig.4. There are two groups of curves in the figure, one for the learning rate $\alpha = 0.05$ and the other for $\alpha = 0.01$. From the figure, we see that the reward curves for $W$ and $Q$ learning will reach the optimal level at around the same number of episodes for the same learning rate but the reward curve for $W$-learning will grow more rapidly. Here, in this experiment, we set the learning rates for different levels to be equal: $\alpha_1 = \alpha_2 = \alpha$. In fact the learning speed is generally determined by the low level learning rate $\alpha_2$ and high level learning rate $\alpha_1$ has less effect.

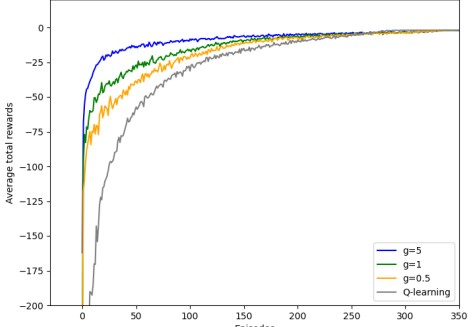 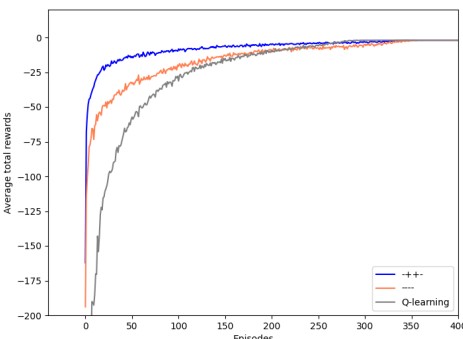

Figure 5: Together with $Q$-learning, the total rewards of $W$ learning are shown on the left in the case of $|g_1| = 0.5$, $|g_1| = 1$ and $|g_1| = 5$. Total rewards for various signs of the parameter $g_1$ with the same absolute value $|g_1| = 5$ are also presented. The reward is plotted by taking the average of 200 runs.

As we have seen, there is an enhancement for the $W$-learning with the choice of update parameters shown in equation 22 and here we will explore more about the effects of the parameters $g_1$. We first choose to change the absolute value of the parameter from 5 to 0.5 and the result is presented on the left of Fig.5, which shows that the bigger absolute value of $g_1$ will tend to lead to better performance. Apart from changing the absolute value of the parameter, we can also vary their signs. For a given

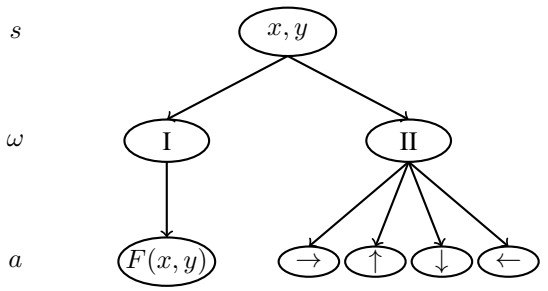

Figure 6: The hierarchical structure is shown on the left figure where the state dependent strategy $F(x, y)$ is illustrated by the figure on the right.

absolute value $|g_1|$, in our case $|g_1| = 5$, there are sixteen kinds of combinations of the parameters which can be sorted into three groups corresponding to three types of curves shown in the right hand side of the Fig.5. Following the convention introduced in appendix C, we label the parameters with a combination of four $\pm$ signs. For example, the case we have studied in equation 22 can be denoted as $(-++-)$. The first group of parameters behaves the same as $(-++-)$, containing $(++--), (+++-)$ and $(+-++)$. The second group follows the curve for $Q$-learning, consisting

of $(+--+), (--++), (---+)$ and $(+-++)$. The third group of curves lies between them which contains $(+---), (-+++), (+-+-), (-+-+), (++-+), (--+-), (++++)$ and $(----)$. All of the details are shown in appendix C.

Now we consider the case where the goal is not located in the corner; therefore we need to introduce a state dependent policy. In our case, the goal is set at $(6,6)$. To construct the strategy, we consider the optimal policy without the obstacles and denote it as $F(x,y)$ then the hierarchical structure of the strategy is shown in Fig.6 where the figure on the left represents the strategy and the right figure shows the detail of the function $F(x,y)$.

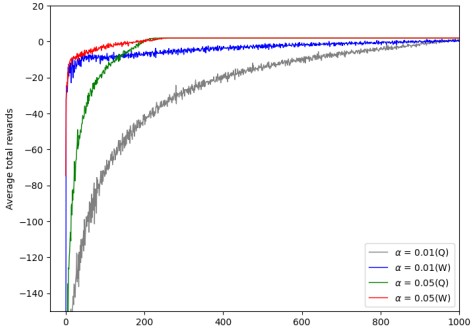 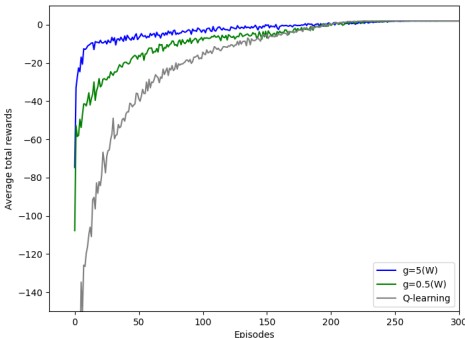

Figure 7: Total rewards for the $W$ and $Q$ learning with learning rate $\alpha = 0.05$ and $\alpha = 0.01$ are shown on the left while the results for $W$ learning is illustrated in the right figure in the case $|g_1| = 5$ and $|g_1| = 0.5$. The reward is plotted by taking the average of 200 runs.

Given the strategy, we first plot the reward for the $Q$ and $W$ learning and put the result together shown on the left hand side of Fig.7. One can see the performance for the $W$ learning is similar to the previous case when the goal is set at $(7,7)$ and $Q$-learning shows slight improvement since the distance between the starting point and the goal (now at $(6,6)$) becomes shorter. Moreover, we also present the results for different update parameters $|g_1| = 0.5$ in the right figure, showing that the agent tends to find out the optimal policy more quickly if the $|g_1|$ takes bigger value.

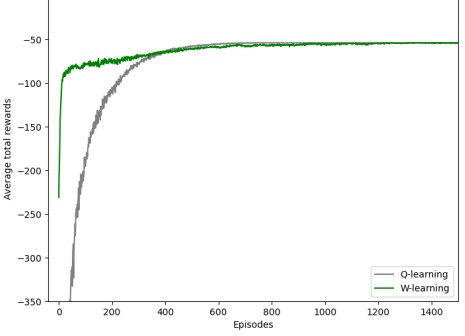 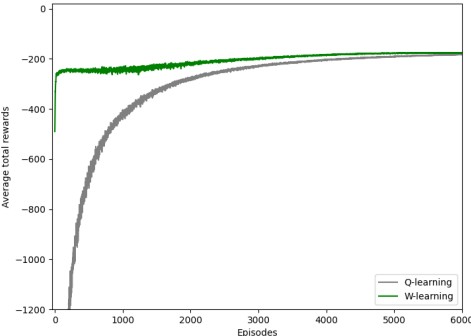

Figure 8: The performance of the training when the size of the grid world is changed. The left is for the 20-by-20 grid world while the right is for the 50-by-50. The reward is plotted by taking the average of 200 runs.

Then we investigate how training behaves as the grid world size increases. The performance for the 20-by-20 grid world is shown on the left and the results for the 50-by-50 one is shown on the right

in Fig.8. The obstacle configurations are the same as the previous 7-by-7 case, so the effect of grid world size becomes dominant. For both cases, the learning rates are set as $\alpha_{20} = \alpha_{50} = 0.5$. From the figure, we observe that, since it generally takes more steps for the agent to reach the goal, the magnitude of the reward will increase for both cases, but the behaviors of $W$ and $Q$ learning differ. For $Q$-learning, as the grid world size increases, the magnitude of the initial reward increases rapidly with the size, but the reward curves approach the optimal one in a similar way. For $W$-learning, the reward tends to stay in a better performance region in initial steps because of the introduction of the hierarchical strategy, which guides the agent to the goal more efficiently. After the early stage, the $W$-learning will enter a more stable phase and then the $W$ and $Q$ learning curves converge to the optimal level roughly at the same time.

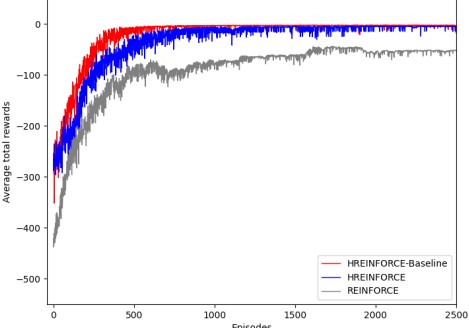 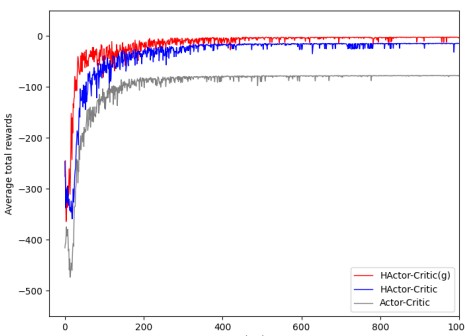

Figure 9: Hierarchical REINFORCE and REINFORCE with Baseline learning results for learning rate $0.005$ are depicted on the left-hand side of the figure. Hierarchical actor-critic learning results are shown on the right-side of the figure with learning rate $0.001$. For HActor-Critic, parameters are set to $G_1 = G_2 = r$ while different update strengths for two levels, $G_1 = g, G_2 = r$, are introduced during the training denoted as HActor-Critic(g). Learning curves are plotted by taking the average of 100 runs and the maximum step for each episode is set as 100.

Based on the study of $W$-learning and $Q$-learning, one can move on to study their generalization to double learning and $n$-step learning. We leave details of the analysis in Appendix D. In the end, we present the study of various learning methods based on the hierarchical policy gradient theorem in Fig.9, including hierarchical REINFORCE, REINFORCE with Baseline and Actor-Critic. Given the plot, one can see that hierarchical learning outperforms its counterpart from two aspects. One is the average convergence speed, and the other is the average convergence limit. The increase for the learning speed results from the introduction of the strategies and the asymptotic reward value reflects the robustness of the training. Here we set the maximum training step for each episode as 100. For the standard learning procedure, if by chance the agent cannot reach the goal within 100 steps, it tends to fail to reach the goal during all the following training episodes. This, on average, contributes to deviation from the optimal value. The agent equipped with a strategy in general takes fewer steps to reach the goal. Fewer failed episodes occur during the training, therefore the reward curve on average converges to the value closer to the optimal one.

## 4 RELATED WORKS

**Option Framework** We can compare the option framework (Sutton et al., 1999b) with $W$-learning contains two layers, where the first level learns the subgoals while the higher level learns the policy over options, corresponding to the option and intra-option Q-learning for a semi-MDP. One of the main differences is that there is no need to define the termination function $\beta$ in the context of $W$-learning. In fact, $W$-learning procedure is more like setting $\beta = 1$ for all states i.e., we choose a new option at every step. The other difference is the details of the update rules for two levels. For $W$-learning, we need to generalize the reward to a series of parameters $g$ at different levels while the update rule for the option framework depends on the termination function. In both cases, these quantities must be fixed by hand before training.

**MAXQ** In the context of MAXQ (Dietterich, 2000), the $Q$ function is divided into the local and global pieces according to the termination function. The parent task is divided into subtasks represented in terms of the sequence of the agent's actions and states. Here the strategy in $W$-learning reflects the agent's reasoning between a single state and action, therefore all the $W_k$ functions are global, taking all the following steps into consideration. Moreover, the optimal policy for MAXQ is determined recursively, which requires all the subtask policies to be optimal. Therefore, the recursively optimal policy is weaker than the globally optimal policy given by $Q$-learning. For $W$-learning, the optimal policy is determined in the same way as $Q$-learning by figuring out the unique fixed point of the Bellman optimality operator. It enables us to find stronger policy compared to the globally optimal policy by constructing more complicated clip structure.

**Option-Critic** The option-critic method (Bacon et al., 2017) is the further development of semi-MDP based on the study of gradient of intra-option policy and termination function. Compared with the option framework, rather than input the termination function before the training, it enables the agent to learn $\beta$ during the options improvement part of the training procedure. A learnable termination function requires fewer input data during the training but there is no evidence that better performance is guaranteed. In the context of $W$-learning, we are able to directly learn intra-option policy and policy over options at the same time with the help of hierarchical gradient theorem while a hand-crafted architecture for the policy is often required if the agent wants to outperform the standard one.

**Strategy Discovery** From the study of projective simulation, we see that there are fruitful structures to investigate, for example, the dynamic evolution of the clips. In the context of hierarchical reinforcement learning, this is connected to the evolution of the strategy, that is, how to train the agent to learn strategies by itself rather than fixing them manually before training. The discovery of strategies is studied in the reinforcement learning literature (McGovern & Barto, 2001; Menache et al., 2002; Şimşek & Barto, 2004; Şimşek et al., 2005; Gregor et al., 2016; Machado et al., 2017).

We are using hand-crafted architecture in this paper so the agent has better performance than the standard one. It is possible to let the agent generate new architectures by itself. For example, in the navigation task, one can start from a fully connected two-level clip network where the two high-level strategies are connected to the four action nodes. Then we put the agent in the empty box without obstacles and do the pretraining. During the pretraining, we can set up a threshold with a small parameter $\epsilon$ and remove the low probability legs for $\pi(a \mid \omega) < \epsilon$. After the pretraining, we fix the architecture and apply it to the more complicated tasks with obstacles. In our case, we start from two identical strategies and they are both connected to all four actions. During the pretraining procedure, we shape the architecture of one strategy to $F(x, y)$ by cutting the lower probability legs out of the network. At the same time, we keep the other strategy network fully connected so that the agent could be flexible to deal with more dynamic tasks involving obstacles. Ultimately, we will obtain the architecture presented before and we can see it makes the agent perform better than the standard agent without the hierarchical structure.

**Deep Learning** Based on the development of $W$-learning, one can consider the hierarchical generalization of more advanced learning algorithms, for example, the area of deep reinforcement learning. Similar to the deep $Q$-network, combining $W$-learning with deep learning will give us a powerful tool to deal with more complicated tasks. Therefore, it will be interesting to see the application of the $W$-learning to the real-life situations and compare the performance with other hierarchical reinforcement learning methods (Hasselt, 2010; Mnih et al., 2015; 2016; Van Hasselt et al., 2016; Kahn et al., 2018; 2021; Lee & Yusuf, 2022; Hu et al., 2024; Gök, 2024; Zhu et al., 2025).

## 5 CONCLUSIONS

In this article, based on the study of projective simulation, we introduce a new method for hierarchical reinforcement learning, which generalizes $Q$-learning to the hierarchical $W$-learning. The $W$-learning enables us to introduce various strategies to the agent, and from the experiment we can see that a proper choice of strategy significantly improves training performance. The main part of this article focuses on the construction of the theoretical framework and we only illustrate the principle of the learning procedure through a navigation task while we should note that the hierarchical $W$-learning method should work on any other task with a strategy that could be represented as a hierarchical clip structure.

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

# A  POLICY GRADIENT

In this appendix, we present the algorithms for the hierarchical REINFORCE, REINFORCE with Baseline and actor-critic.

---

**Algorithm 2**: Hierarchical REINFORCE

---

Input: a set of differentiable policy parameterization $\pi(a \mid \omega_l), \pi(\omega_{i+1} \mid \omega_i)$ and $\pi(\omega_1 \mid s)$
Algorithm parameters: step size $\alpha_i > 0$
Initialize policy parameters $\theta_i$

Loop forever (for each episode):
 Generate an episode $S_0, \Omega_0, A_0, R_1, \ldots, S_{T-1}, A_{T-1}, R_T$, following $\pi_{\theta_i}(\cdot|\cdot)$
 Loop for each step of the episode $t = 0, 1, \ldots, T - 1$
  $G \leftarrow \sum_{k=t+1}^{T} \gamma^{k-t-1} R_k$
  $\theta_i \leftarrow \theta_i + \alpha \gamma^t G \nabla \ln \pi_{\theta_i}(\Omega_{i+1} \mid \Omega_i)$

---

**Algorithm 3**: Hierarchical REINFORCE with Baseline

---

Input: a set of differentiable policy parameterization $\pi(a \mid \omega_l), \pi(\omega_{i+1} \mid \omega_i)$ and $\pi(\omega_1 \mid s)$
Input: a set of strategy-value function parameterization $\hat{u}_{w_i}(s)$
Algorithm parameters: step size $\alpha_i^\theta > 0, \alpha_i^w > 0$
Initialize policy parameters $\theta_i, w_i$

Loop forever (for each episode):
 Generate an episode $S_0, \Omega_0, A_0, R_1, \ldots, S_{T-1}, A_{T-1}, R_T$, following $\pi_{\theta_i}(\cdot|\cdot)$
 Loop for each step of the episode $t = 0, 1, \ldots, T - 1$ and for all $i$
  $G \leftarrow \sum_{k=t+1}^{T} \gamma^{k-t-1} R_k$
  $\delta_i \leftarrow G - \hat{u}_{w_i}(s)$
  $w_i \leftarrow w_i + \alpha_i^w \delta_i \nabla \hat{u}_{\omega_i}(S_t)$
  $\theta_i \leftarrow \theta_i + \alpha_i^\theta \gamma^t \delta_i \nabla \ln \pi_{\theta_i}(\Omega_{i+1} \mid \Omega_i)$

---

**Algorithm 4**: Hierarchical Actor-Critic

---

Input: a set of differentiable policy parameterization $\pi(a \mid \omega_l), \pi(\omega_{i+1} \mid \omega_i)$ and $\pi(\omega_1 \mid s)$
Input: a set of strategy-value function parameterization $\hat{u}_{w_i}(s, \omega_i)$
Algorithm parameters: step size $\alpha_i^\theta > 0, \alpha_i^w > 0$, update strength $G_i$
Initialize policy parameters $\theta_i, w_i$

Loop forever (for each episode):
 Initialize $S$
 $I \leftarrow 1$
 Loop while $S$ is not terminal (for each time step ):
  $\Omega_1 \sim \pi(\cdot \mid S), \ldots, \Omega_i \sim \pi(\cdot \mid |\Omega_{i-1}), \ldots, A \sim \pi(\cdot \mid \Omega_{l-1})$
  Take action $A$, observe $S', G_i, R$
  $\delta_i = G_i + \gamma \hat{u}_{w_i}(S') - \hat{u}_{w_i}(s)$
  $w_i \leftarrow w_i + \alpha_i^w \delta_i \nabla \hat{u}_{w_i}(S)$
  $\theta_i \leftarrow \theta_i + \alpha_i^\theta I \nabla \ln \pi_{\theta_i}(\Omega_{i+1} \mid \Omega_i)$
$I \leftarrow \gamma I$
$S \leftarrow S'$

---

## B CONVERGENCE

In this section, we will show the proof of the convergence for the update rules equation 9 and equation 18. The convergence for updating the $W$ function is guaranteed by the following theorem.

**Theorem 2** *Suppose that the learning rate $\alpha_i(t)$ are nonnegative and satisfy*

$$\sum_{t=0}^{\infty} \alpha_i(t) = \infty, \qquad \sum_t^{\infty} \alpha_i^2(t) < \infty \qquad \forall\, i. \tag{23}$$

*Then $W(s, \omega_1, \ldots, \omega_i)$, for all $i$, converge with probability 1 to $W^*(s, \omega_1, \ldots, \omega_i)$ provided that all policies are proper.*

To prove that the $W$ function will converge to the optimal one $W^*$ following the update rule. We first define the transition probability involving the strategies $p(s', \omega_1', \ldots \omega_{i-1}' \mid s, \omega_1, \ldots, \omega_i)$ as

$$p(s', \omega_1', \ldots, \omega_{i-1}' \mid s, \omega_1, \ldots, \omega_i) = \sum_{s,a,\omega_{i+1},\ldots,\omega_{l-1}} \pi(\omega_{i-1}' \mid \omega_{i-2}) \cdots \pi(\omega_1' \mid s') \tag{24}$$

$$\times p_{s's}(a) \pi(a \mid \omega_{l-1}) \cdots \pi(\omega_{i+1} \mid \omega_i) \cdots \pi(\omega_1 \mid s)$$

where $p_{s's}(a)$ is the probability that the agent transforms to the state $s'$ given the previous state $s$ and the action $a$. By definition, we say that the policy is proper when there exists a function $\xi(s, \omega_1 \ldots \omega_{i-1})$ such that the condition

$$\sum_{s'\omega'} p(s', \omega_1', \ldots, \omega_{i-1}' \mid s, \omega_1, \ldots, \omega_i) \xi(s', \omega_1', \ldots \omega_{i-1}') \leq \beta \xi(s, \omega_1, \ldots, \omega_{i-1}) \tag{25}$$

is satisfied for all $s, \omega_1, \ldots, \omega_i$ and $\beta \in [0, 1)$. Moreover, we define the weighted maximum norm of the $W$ function as

$$\|W(s, \omega_1, \ldots, \omega_i)\|_\xi = \max_{s,\omega_1,\ldots,\omega_i} \frac{|W(s, \omega_1, \ldots, \omega_i)|}{\xi(s, \omega_1, \ldots, \omega_{i-1})} \tag{26}$$

and the mapping $H$ from one $W$ function to the other as

$$HW(s, \omega_1, \ldots, \omega_i) = g + \sum_{s'\omega'} p(s', \omega_1', \ldots, \omega_{i-1}' \mid s, \omega_1, \ldots \omega_i) \max_{\omega'} W(s', \omega_1', \ldots, \omega_{i-1}', \omega'). $$

Given the above definition, the update rule for the $W$ function can be written as

$$W_{t+1} = (1 - \alpha_t)W_t + \alpha_t(HW_t + \eta_t) \tag{27}$$

where $\eta_t$ is the noise term given by

$$\eta_t(s, \omega) = g + \max_{\omega_i'} W_t(s', \omega_1', \ldots, \omega_{i-1}', \omega_i') - HW_t(s, \omega). \tag{28}$$

Moreover, given the update history $\mathcal{F}_t^i = \{s_0, \ldots, s_t, \omega_0, \ldots, \omega_t^i, a_0, \ldots, a_{t-1}\}$, one can deduce that

$$\mathbb{E}[\eta_t \mid \mathcal{F}_t^i] = 0 \qquad \mathbb{E}[\eta_t^2 \mid \mathcal{F}_t^i] \leq C, \tag{29}$$

where $C$ is a constant number. In our case, $s = s_t$ and $s' = s_{t+1}$. Now, we consider the difference between two functions $W$ and $\bar{W}$. Acting the transformation $H$ on it, we have

$$|(HW - H\bar{W})(s, \omega)| \leq \sum_{s'\omega'} p(s', \omega_1', \ldots, \omega_i' \mid s, \omega)| \max_{\omega_i'} W(s', \omega_1', \ldots, \omega_{i-1}', \omega_i')$$

$$- \max_{\omega_i'} \bar{W}(s', \omega_1', \ldots, \omega_{i-1}', \omega_i')| \tag{30}$$

$$\leq \sum_{s'\omega'} p(s', \omega_1', \ldots, \omega_{i-1}' \mid s, \omega)| \max_{\omega_i'} |W(s', \omega_1', \ldots, \omega_{i-1}', \omega_i')$$

$$- \bar{W}(s', \omega_1', \ldots, \omega_{i-1}', \omega_i')| \tag{31}$$

$$\leq \sum_{s'\omega'} p(s', \omega_i', \ldots, \omega_{i-1}' \mid s, \omega) \xi(s', \omega_i', \ldots, \omega_{i-1}')\|W - \bar{W}\|_\xi.$$

where we are using $\omega = (\omega_1, \ldots, \omega_i)$ and $\omega' = (\omega'_1, \ldots, \omega'_{i-1})$ for simplicity. Given that the policy is proper, we obtain the relation

$$||(HW - H\bar{W})(s, \omega_1, \ldots, \omega_i)||_\xi \leq \beta ||(W - \bar{W})(s, \omega_1, \ldots, \omega_i)||_\xi, \tag{32}$$

for $0 \leq i \leq l$, therefore we can see that $H$ is a contraction mapping and it converges to a unique value $W^*$ called fixed point of the contraction, i.e. $HW^* = W^*$ Luenberger (1997); Bertsekas & Tsitsiklis (2015).

Together with the condition equation 29, we then prove the convergence of the $W$ function according to the convergence property for the random iterative process shown in Jaakkola et al. (1993); Bertsekas & Tsitsiklis (1996). The main idea of the proof for the convergence is to show that $||W_t - W^*|| \leq D_t$ for some value $D_t$ at each step and then to verify $D_t \to 0$ when $t$ goes to infinity. This is easy to see when $\eta_t = 0$ since we have $|HW_t - W^*| \leq \beta |W_t - W^*|$ while more sophisticated construction of $D_t$ is required when $\eta_t \neq 0$. For details, see the section 4.3.6 of Bertsekas & Tsitsiklis (1996).

The proof of the convergence for the gradient policy method in equation 18 is the direct application of the Proposition 3.5 shown in Bertsekas & Tsitsiklis (1996) to each $i$, therefore, we have the following theorem.

**Theorem 3** *Suppose that*

$$\max_{\theta_i, j, k} \left| \frac{\partial^2 \pi(\omega_{i+1} \mid \omega_i)}{\partial \theta_i^j \partial \theta_i^k} \right| < L < \infty \qquad \forall\, i, \tag{33}$$

*where $L$ is a constant upper bound and the nonnegative learning rates satisfy the relation*

$$\alpha_i(t) \to 0, \qquad \sum_t^\infty \alpha_i(t) = \infty \qquad \forall\, i. \tag{34}$$

*Then $\pi(\omega_{i+1} \mid \omega_i)$, for all $i$, converge with probability 1 to $\pi^*$ i.e., $\frac{\partial V}{\partial \theta_i} \to 0$ when $t \to \infty$.*

## C  SIGNS

In this appendix, we show the performance for various signs of the parameter $g_2$ when the absolute value is fixed $|g_2| = 5$. For simplicity, the value

$$g_2(\omega_1 = \mathrm{I}, r = -10) = \mp 5 \qquad g_2(\omega_1 = \mathrm{I}, r = -2) = \pm 5 \tag{35}$$
$$g_2(\omega_1 = \mathrm{II}, r = -10) = \pm 5 \qquad g_2(\omega_1 = \mathrm{II}, r = -2) = \mp 5 \tag{36}$$

are denoted as $(- + + -)$ and $(+ - - +)$ therefore, one should note that $(- + + -)$ represents the value chosen in the Fig.3 and the main part of this article. Their performance is shown in Fig.10.

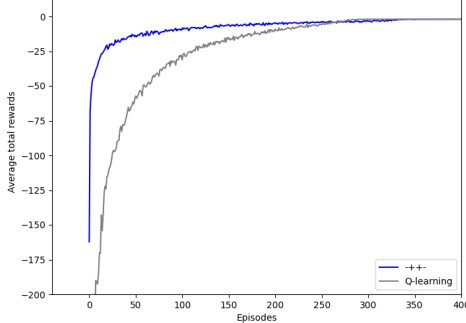 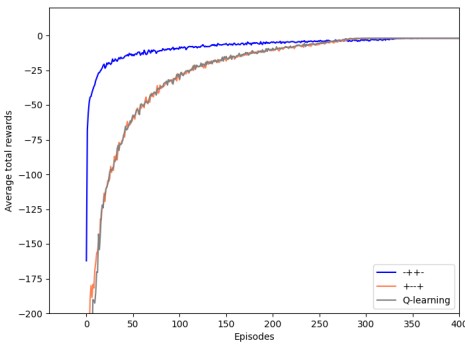

Figure 10: The performance of the training with $g_2$ taking the value of $(- + + -)$ on the left and $(+ - - +)$ on the right. The reward is plotted by taking the average of 200 runs.

We denote the case where

$$g_2(\omega_1 = \mathrm{I}, r = -10) = \pm 5 \qquad g_2(\omega_1 = \mathrm{I}, r = -2) = \pm 5 \tag{37}$$
$$g_2(\omega_1 = \mathrm{II}, r = -10) = \mp 5 \qquad g_2(\omega_1 = \mathrm{II}, r = -2) = \mp 5 \tag{38}$$

as $(+ + --)$, $(- - ++)$ and the plot is shown in Fig.11.

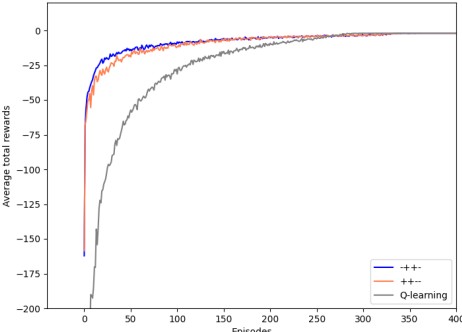 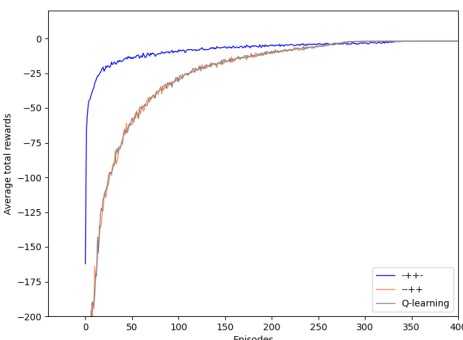

Figure 11: The performance of the training with $g_2$ taking the value of $(+ + --)$ on the left and $(- - ++)$ on the right. The reward is plotted by taking the average of 200 runs.

The value for

$$g_2(\omega_1 = \mathrm{I}, r = -10) = \pm 5 \qquad g_2(\omega_1 = \mathrm{I}, r = -2) = \mp 5 \tag{39}$$
$$g_2(\omega_1 = \mathrm{II}, r = -10) = \mp 5 \qquad g_2(\omega_1 = \mathrm{II}, r = -2) = \mp 5 \tag{40}$$

are denoted as $(+ - --)$ and $(- + ++)$ with the result shown in Fig. 12.

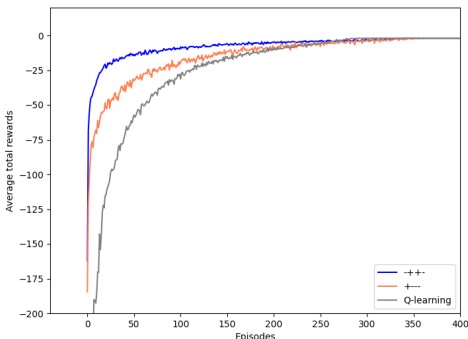 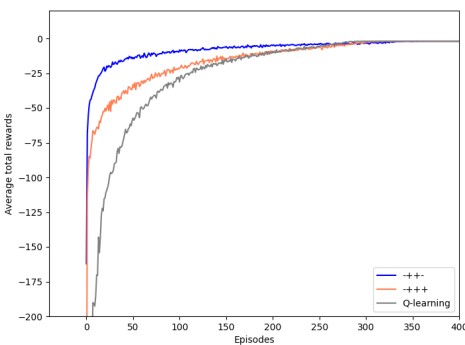

Figure 12: The performance of the training with $g_2$ taking the value of $(+ - --)$ on the left and $(- + ++)$ on the right. The reward is plotted by taking the average of 200 runs.

The value for

$$g_2(\omega_1 = \mathrm{I}, r = -10) = \pm 5 \qquad g_2(\omega_1 = \mathrm{I}, r = -2) = \mp 5 \tag{41}$$
$$g_2(\omega_1 = \mathrm{II}, r = -10) = \pm 5 \qquad g_2(\omega_1 = \mathrm{II}, r = -2) = \mp 5 \tag{42}$$

are denoted as $(+ - +-)$, $(- + -+)$ and the results are presented in Fig.13.

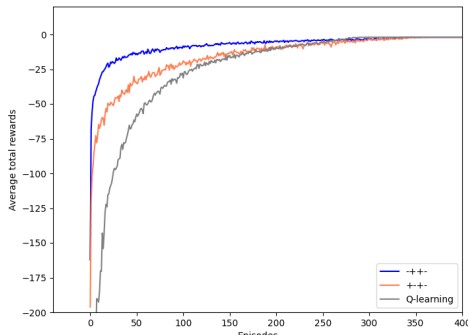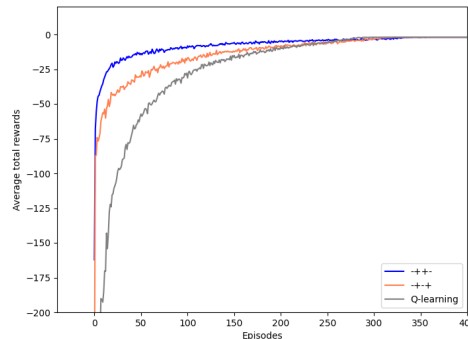

Figure 13: The performance of the training with $g_2$ taking the value of $(+ - + -)$ on the left and $(- + - +)$ on the right. The reward is plotted by taking the average of 200 runs.

The value for

$$g_2(\omega_1 = \mathrm{I}, r = -10) = \pm 5 \qquad g_2(\omega_1 = \mathrm{I}, r = -2) = \pm 5 \qquad (43)$$
$$g_2(\omega_1 = \mathrm{II}, r = -10) = \pm 5 \qquad g_2(\omega_1 = \mathrm{II}, r = -2) = \mp 5 \qquad (44)$$

are denoted as $(+ + + -)$ and $(- - - +)$ with the performance illustrated in Fig.14.

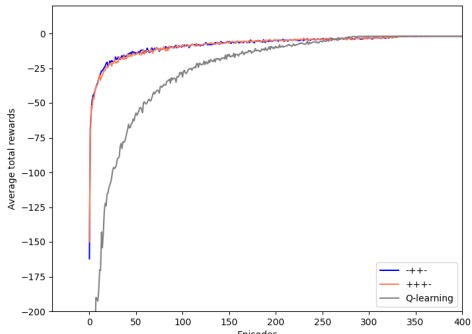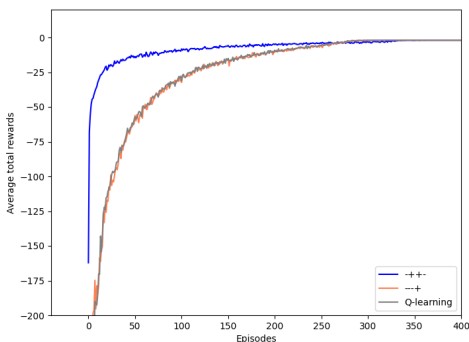

Figure 14: The performance of the training with $g_2$ taking the value of $(+ + + -)$ on the left and $(- - - +)$ on the right. The reward is plotted by taking the average of 200 runs.

The value for

$$g_2(\omega_1 = \mathrm{I}, r = -10) = \pm 5 \qquad g_2(\omega_1 = \mathrm{I}, r = -2) = \mp 5 \qquad (45)$$
$$g_2(\omega_1 = \mathrm{II}, r = -10) = \pm 5 \qquad g_2(\omega_1 = \mathrm{II}, r = -2) = \pm 5 \qquad (46)$$

are denoted as $(+ - + +)$, $(- + - -)$ and the performance is presented in Fig.15.

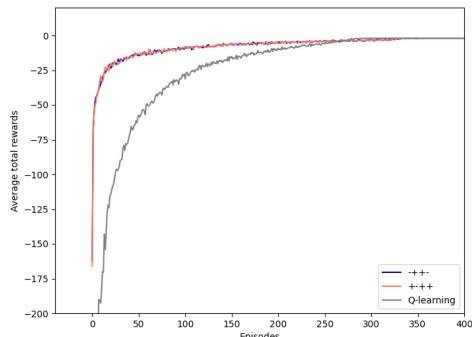 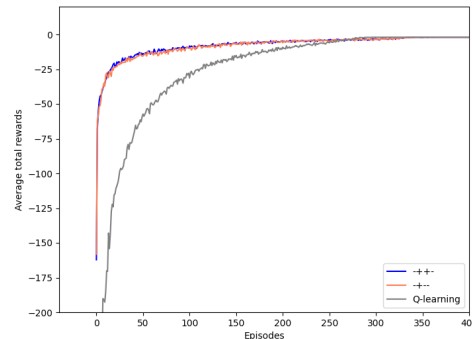

Figure 15: The performance of the training with $g_2$ taking the value of $(+-++)$ on the left and $(-+--)$ on the right. The reward is plotted by taking the average of 200 runs.

The value for

$$g_2(\omega_1 = \mathrm{I}, r = -10) = \pm 5 \qquad g_2(\omega_1 = \mathrm{I}, r = -2) = \pm 5 \qquad (47)$$
$$g_2(\omega_1 = \mathrm{II}, r = -10) = \pm 5 \qquad g_2(\omega_1 = \mathrm{II}, r = -2) = \pm 5 \qquad (48)$$

are denoted as $(++++)$, $(----)$ and the performance is shown in Fig.16.

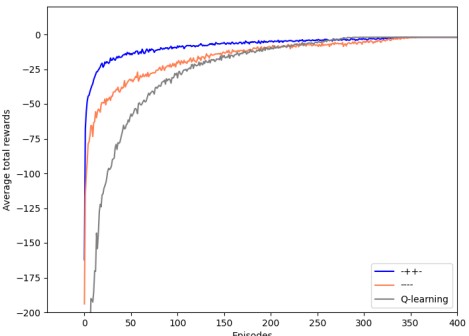 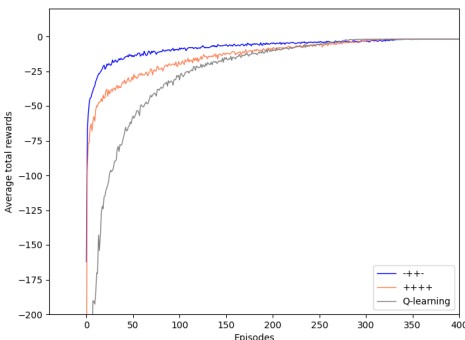

Figure 16: The performance of the training with $g_2$ taking the value of $(----)$ on the left and $(++++)$ on the right. The reward is plotted by taking the average of 200 runs.

The case where

$$g_2(\omega_1 = \mathrm{I}, r = -10) = \pm 5 \qquad g_2(\omega_1 = \mathrm{I}, r = -2) = \pm 5 \qquad (49)$$
$$g_2(\omega_1 = \mathrm{II}, r = -10) = \mp 5 \qquad g_2(\omega_1 = \mathrm{II}, r = -2) = \pm 5 \qquad (50)$$

are denoted as $(++-+)$ and $(--+-)$ with the results shown in Fig.17.

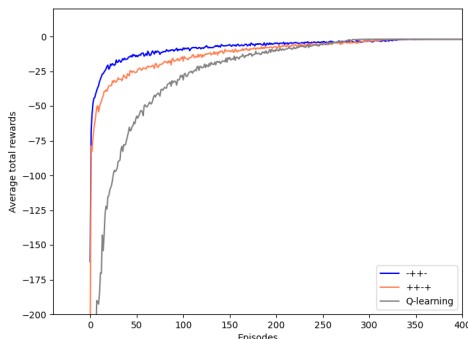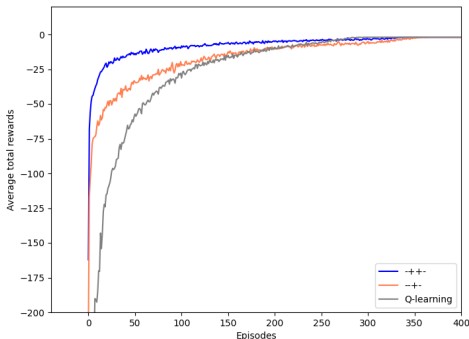

Figure 17: The performance of the training with $g_2$ taking the value of $(+ + -+)$ on the left and $(- - +-)$ on the right. The reward is plotted by taking the average of 200 runs.

## D  DOUBLE W AND N-STEP W-LEARNING

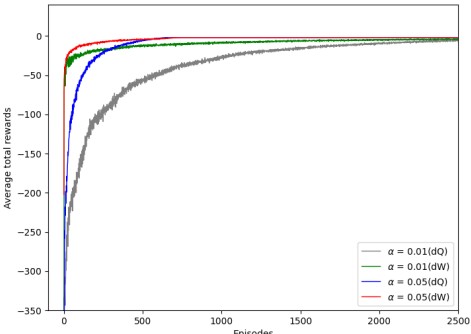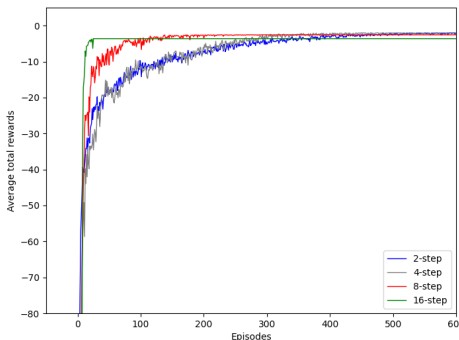

Figure 18: Double Q and W learning results for $\alpha = 0.01$, $\alpha = 0.05$ are shown on the left while the 2, 4, 8, 16-step learning results for W-learning are shown on the right. The reward is plotted by taking the average of 200 runs.

In this section, we present the results of the double learning and $n$-step learning for our original 7-by-7 gridworld task in Fig.18, from which we can see that double W-learning tends to learn faster than double Q-learning under the same learning rate. The results for 2, 4, 8, 16 - step W-learning are presented on the right with learning rate $\alpha = 0.05$. From the figure we can see that there is no significant improvement when the learning step is small, for example, the 2-step and 4-step learning. When the learning step becomes large, the learning speed will increase while the performance will be less robust.

Furthermore, we present the 2, 4, 8, 16-step Q-learning and W-learning results. For the 2-step case, we can see that there is an increase in learning speed compared with the 1-step Q-learning, and there is no significant improvement for the $W$-learning. But 2-step W-learning still outperforms 2-step Q-learning. For the 4-step learning case, there is continuous improvement for the 4-step Q-learning and still no improvement for the 4-step $W$-learning. The learning speed of 4-step Q and W-learning becomes similar.

For the 8-step case, both of the 8-step Q and W-learning experience a speedup and their performance becomes similar. For the 16-step case, there is an increase in learning speed for 16-step Q and W-learning but we can see that Q-learning becomes less stable than the W-learning. In general, when

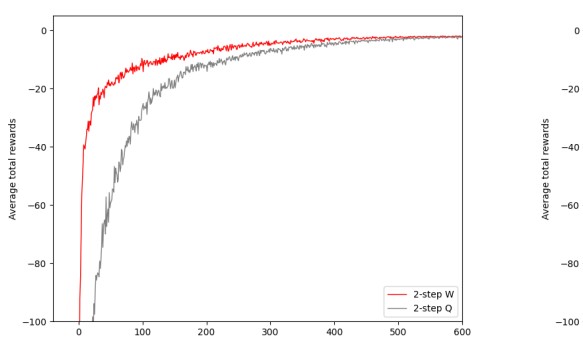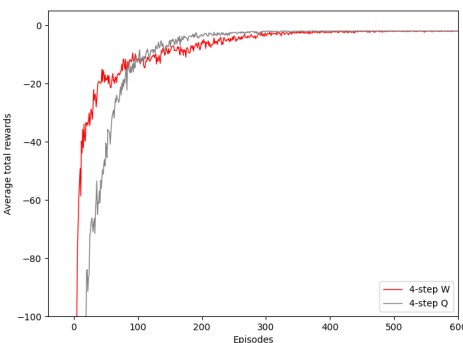

Figure 19: The 2-step and 4-step Q and W-learning results are shown in the figure, in which W-learning is represented by the red curves and Q-learning is represented by the gray ones. The reward is plotted by taking the average of 200 runs.

the number of steps is increased during training, more episodes occur in which the agent fails to identify the optimal policy. Overall, $W$-learning appears more robust than Q-learning.

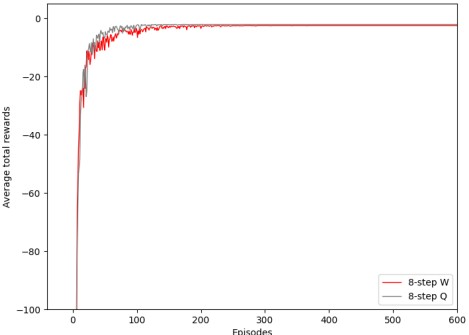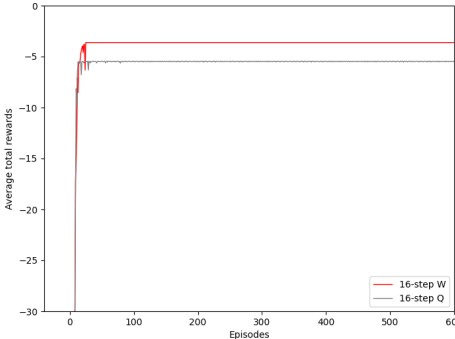

Figure 20: The 8-step and 16-step Q and W-learning results are shown in the figure, in which W-learning is represented by the red curves and Q-learning is represented by the gray ones. The reward is plotted by taking the average of 200 runs.

