# OpenReview forum: "Hierarchical W-Learning"
_ICLR.cc/2026/Conference — Submitted to ICLR 2026_

### Official Review · Reviewer_GJFn · 2025-10-18

**Soundness:** 1
**Presentation:** 2
**Contribution:** 1
**Rating:** 2
**Confidence:** 4

**Summary:**

The authors claim to „develop a simple and generic new method for hierarchical reinforcement learning in this article“.

**Strengths:**

n/a

**Weaknesses:**

The experimental evaluation is insufficient. Just a very  simple (scalable) grid world problem is considered.

More importantly, the text does not study a new RL approach in isolation.
The experiments mainly compare two settings: RL without prior knowledge about the task and RL with prior knowledge. Any reasonable algorithm that incorporates meaningful prior knowledge should perform better, right?

It is not about hierarchical vs non-hierarchical, because all „hierarchical“ methods have more information about the task than the standard methods.

In the experiments, the „W-learning“ method is given extra information about the task.
See Figure 3, left branch, which gives hints for a target in the upper right corner (as it is the case in the first experiments) or, explicitly, the definition of F, the optimal policy w/o obstacle, in line 380. The function is provide in the „W-learning“ experiments, see Fog. 6.

There are many control experiments that are not considered, for example:
1. Simplest one: Start standard Q-learning with the Q function initialised to the optimal policy without obstacles (i.e. start with Q^F in a table).
2. Starting from an agnostic policy pi with parameter vector theta, define the policy pi’(s) = \theta’ F(s) + (1-theta’) pi(s) that incorporate prior information and use some actor-critic method to learn pi’ , i.e., learn (theta, theta’) .

Another comment regarding the presented experiments/results:
As the objective functions are different, the learning rates could be significantly different for Q- and „W“-learning.
While it is appreciated that several learning rates were tested, it would be interesting to see the results for an even higher learning rate for Q-learning (which is perhaps cooled down over time). Would there still be a significant difference?
What about a different initialisations of the Q function?



## Minor:

The factorisation in (3) seems to be only for a chain, not for a general hierarchical model.

„and many people believe that building a proper model of it may finally lead to artificial intelligence.“: This is a completely empty statement w/o giving a (working) definition of AI.

„As we can see, the clip structure is a model for the neural network in the brain.“: No, I do not see this. The figure shows some generic acyclic graph. This is totally empty statement. That there is no recurrence in the figure makes it extra sad.

What have the path integrals in quantum mechanics to do with the study?

The authors could check out the literature on policy/state-space factorisation.

Who is "et al." in the reference "Richard S Sutton, Andrew G Barto, et al. Reinforcement learning: An introduction, volume 1. MIT
press Cambridge, 1998."?

**Questions:**

See questions in "Weaknesses" above.

---

> ### Author Response · Authors · 2025-11-19
>
> Thanks for the insightful feedback.
>
> I agree that we are feeding extra information to the agent, so it performs better. But in the context of hierarchical reinforcement learning, I think that is the standard approach. I wrote a new section at page 9 and 10 comparing our work with others. (I really should do this before.)
>
> I can decompose the current state of hierarchical learning into two levels. The first is to study how to feed the information to the agent so it could perform better and execute hard tasks. The second is how to make the agent grow up by itself so that it has better performance in the task. Achieving the second goal is appealing while I do not think the first goal is meaningless. Hierarchical reinforcement learning has been developed for long time while the first goal is not reached yet. If we can teach students some skills so that they can get better scores in the exam, then, why not?
>
> I think the question is really how to tell a strategy to the agent and our approach is simpler and could be extended to a large class of problems in principle. I also wrote some discussions on the strategy discovery at page 10, As you said, start from zero and make the agent evolve by itself. But the method described there is still quite intuitive.
>
> For the experiment, all of our curves are plotted by taking the average of 200 or 100 runs with random initial Q function. So I think it is convincible enough to support our arguments. For higher learning rate (around 1), I think the step is so big that agent tends to fail to identify the optimal point.
>
> For Minor: For equation 3, I think it is better to read it from right to the left. It is more like we are constructing or defining a policy rather than factorizing a known policy.
>
> It is a subtle question to discuss whether an AI model should be based on the biological study of brain. I will leave this question open.
>
> The PS model is indeed a model for the brain (though not well known) as you can see from literatures we have cited, which belongs to the area of quantum reinforcement learning. I think there is already a significant amount of work on this topic. And there is strong connection with the path integral. From the percept to the actuator, there are various of paths to choose and they are weighed by the probability. Similar to the superposition of  paths for a particle, going from initial state and ending at final state. By taking the continuous limit of those discrete clips, (we expect) the mathematical structure should be the same.
>
> We are not doing factorization here, we are building a deeper hierarchical network. I am wondering if you can give me some references in your mind so I can explain more on this point.
>
> Citation typo on the authors corrected.

---

### Official Review · Reviewer_6yDF · 2025-10-23

**Soundness:** 2
**Presentation:** 2
**Contribution:** 2
**Rating:** 2
**Confidence:** 4

**Summary:**

The paper proposes a hierarchical reinforcement learning approach called W-learning, The method is based on abstractions called "Clips" which are intermediate structures or goals between the perception and the action. Rather than learning to map perceptions to actions, the execution engine goes through a sequence of clips until the final clit selects the action. Q-learning is generalized to W-learning, where W is a function of state, strategy and action. The approach is illustrated in two grid navigation domains where the intermediate "clips" correspond to  the task of reaching the final destination and the task of avoiding obstacles. The hierarchical approach is shown to perform better than the non-hierarchical approach.

**Strengths:**

Hierarchical reinforcement learning is a well-studied topic and yet not fully understood. The paper addresses an interesting problem and presents an apparently successful approach. The empirical results are reasonable.

**Weaknesses:**

The paper does not define the terms precisely enough to evaluate the soundness of the approach. Words like "Clip" and "strategy" need to be more precisely defined. It cites most relevant work, but does not explain how this work is different. For example the options or MAXQ framework addresses hierarchical RL in a much more precise notation and correctness proofs. The current framework is different, but it does not explain the embedded assumptions carefully and does not give the proof of the main theorem.

To illustrate in a problem, it is not clear how one should interpret Figure1. The clips do not sequentially follow each other, but look like a directed acyclic graph. It is said that the clip structure is a "model of the brain". But what is its connection to the strategy hierarchy? Why is the brain relevant to model here? As a contrasting example, MAXQ is organized as a graph structure over tasks, where tasks have termination conditions, and are similar to subroutines. The procedural semantics of the MAXQ hierarchy allows a sound derivation of global value function in terms of local value functions. While the paper seems to be attempting a similar decomposition, it is much less clear because the strategies here are not tasks and they do not have termination conditions.

Appendices A and B are missing.

It is also not clear what notion of optimality is applicable, since there are in general many notions (recursive optimality, hierarchical optimality, global optimality) might be in play.  It appears that V(s) should be a function of the strategy, but the paper does not acknowledge that. The meanings of different W functions should be clearly stated.

Hierarchical policy gradient theorem: Can someone view the hierarchical policy controlled by a set of parameters as simply a policy that takes states and policy parameters and outputs a primitive action. If so, then can't someone just use the policy gradient theorem?

The grid domains are too weak to illustrate the power of the framework. The domains seem to be setup such that each primitive action falls into one of the other goals. What happens if some actions fall under both goals, i.e, avoiding obstacles and also moving towards the goal. More ambitious domains have been attempted in the past literature on hierarchical RL. Multiple levels of hierarchies would be more interesting.

**Questions:**

Question: Clearly define the framework. What is a strategy? How does the hierarchical policy work given a set of parameters? Can hierarchical policy gradient can be viewed as an instance of simple policy gradient with a new (hierarchical) parameterization.

---

> ### Author Response · Authors · 2025-11-19
>
> Thanks for the detail comments.
>
> Interesting question! Introduction of strategies is more than introducing parameters to the policy. As you can see from the clip network, strategies are nodes in the network and there are transitions probabilities flowing between different levels. It is more like we are making the decision procedure deeper.
>
> Then l think you may ask, what is the difference between making the parameters deeper (for example deep reinforcement learning) and making the policy deeper. I think these two are parallel concepts. Using deep learning to make the parameter deeper makes the agent more adaptable to various data while introducing strategies and making the policy deeper makes the agent more capable of dealing difficult tasks. It is possible to make both deeper and you can see my discussions at the end of section Related Works at page 10.
>
> For the weakness. We add a new section at page 9 and 10 to discuss the differences between our methods and other frameworks. Also, more clarification about the strategy and clip network is discussed at page 6 around line 285. The rigorous proof for the hierarchical policy gradient theorem is presented before it from equation 12 to equation 15.
>
> I put appendixes A-D in the supplementary section. It should be fine this time since I have put them together with the main part.
>
> We are using global optimality during the whole work. V is not a function of strategies while W are functions of strategies. Moreover, given the hierarchical structure, strategies at different levels correspond to different W functions, we label them by W_k.
>
> For the parameterization problem, see the previous part about the question.
>
> At each step, the agent will choose one of strategies, and based on the chosen strategy, the agent chooses the strategy for the next level or determine the action to execute. So, the agent will not fall into two goals at the same time.

---

> > ### Comment · Reviewer_6yDF · 2025-11-27
> >
> > I thank the authors for their response.
> >
> > Unfortunately the responses were not sufficient to address my concerns.For example, these two responses seem to contradict each other:
> >
> > (1) "We are using global optimality during the whole work."
> >
> > (2) "At each step, the agent will choose one of strategies, and based on the chosen strategy, the agent chooses the strategy for the next level or determine the action to execute."
> >
> > The second notion gives rise to "recursive optimality" according to Dietterich's MAXQ paper. The globally optimal policy may not be expressible as a hierarchy, and even the best policy that can be expressed as a hierarchy (hierarchically optimal policy) may not be learnable recursively.
> >
> > Overall, the claims of the paper are too imprecise and not well-defended. The paper needs to make stronger connection to prior work, tighten the theoretical claims, and evaluate the approach in more challenging domains.
> >
> > My reviews and the scores remain the same.

---

> > > ### Author Response · Authors · 2025-11-28
> > > **Clarifiaction on optimality**
> > >
> > > Thanks for the deeper feedback.
> > >
> > > There is no contradiction between (1), and (2). We are not trying to find the optimal policy recursively. In fact, the policy we found, at least in the navigation task, is even stronger than the globally optimal policy (If one defines the globally optimal policy as the policy induced by the fixed point of the Bellman optimality operator for Q function).
> > >
> > > The reason is that we are trying to find the fixed point of our W function, which could be designed more complicated than Q function by proper choice of \omega. That is exactly what happens in our navigation task where strategy II is the standard learning procedure. Finding our subgoal optimal policy for II is equivalent to finding the globally optimal policy for Q-learning. Again, I should emphasize that we are not trying to identify the optimal policy recursively. We are trying to present an update rule for W function, which is parallel to the Q-learning, so that the unique fixed point for W is identified. The reason that we need to update W functions layer by layer is that we need to determine all intermediate strategies leading to the optimal policy.
> > >
> > > If one worries that the optimal policy figured out by updating W function is weaker than the policy determined by Q function, one can always set the standard learning as one of strategies in W-learning. In such case, the optimal policy determined by the W-learning is always equally strong or stronger.
> > >
> > > I hope this will be helpful to clear some of the confusion.

---

### Official Review · Reviewer_8g86 · 2025-11-01

**Soundness:** 2
**Presentation:** 3
**Contribution:** 2
**Rating:** 4
**Confidence:** 4

**Summary:**

This paper proposes a method for hierarchically learning the Q-function in reinforcement learning (RL), inspired by projective simulation. A new W-function is introduced, which incorporates both the state sss and a hierarchical strategy $\omega$. The authors derive corresponding policy gradient theorems and present an empirical experiment on a navigation task demonstrating that the proposed method achieves faster convergence compared with standard Q-learning.

**Strengths:**

* The idea of incorporating a strategy component, in addition to actions, into the RL learning process is interesting and establishes a meaningful connection to hierarchical RL.
* The paper provides a detailed and illustrative navigation task that effectively bridges theoretical definitions (e.g.,  $\omega$ and $g$) with practical scenarios.

**Weaknesses:**

Motivation: The hierarchical modeling of the Q-function is not well motivated. Although the authors briefly mention “modeling intra-option behavior” and “handling dynamic tasks,” these ideas are neither theoretically developed nor empirically verified.

Experimental validation: The experiments are insufficient to convincingly demonstrate the advantages of W-learning over existing methods such as Q-learning.

- Evaluation is limited to a single navigation task, with comparison only against Q-learning.
- Even within this prototype task, W-learning achieves similar asymptotic performance to Q-learning, differing mainly in faster convergence speed.

Clarity of definitions: The relationship between the strategy $\omega$, action $a$, and W-function could be more clearly defined.

- Based on the paper’s description, ω\omegaω appears to act as an internal or intermediate action, and the W-function seems analogous to a Q-function extended with this additional variable. However, in the navigation task example, the strategy behaves more like a subgoal, which could be more explicitly articulated.

Minor:
The paper should use `\citep{}` instead of `\cite{}` for most references.

**Questions:**

How should one design and map the strategy variable $\omega$ in practice? Since this appears to be highly task-specific, does the initialization of  $\omega$ significantly affect performance, and how might this approach scale to more complex tasks with many possible actions and strategies?

---

> ### Author Response · Authors · 2025-11-19
>
> Thanks for the useful feedback.
>
> For experiment: Two more experiments on REINFORCE and REINFORCE with Baseline are added. It is interesting to see that the learning curves differ from both the convergence speed and convergence limit point of view. Results and discussions are added at page 9.
>
> For clarification: To be honest, I think there is no significant difference between the action and strategy except for the fact that the action directly leads to agent’s execution. In terms of the clip network, state and action are the initial and end nodes while strategies are the internal nodes. We have added more illustration on this point at page 6 (line 285-line 290).
>
> For Minor: It seems that the citation looks better if we use \citep{}. (not sure)
>
> For questions. Yes, an arbitrary strategy structure should work for all tasks due to the convergence theorem, but we need to properly design the strategy in order to obtain better performance comparing with the agent without a strategy. This is highly task-specific. Till now, I think there is no standard method for strategy discovery while I add some discussions (at page 10) on this in the new section Related Work. By cutting off low probability legs during the pretraining, one can get the strategy shown in our paper.

---

> > ### Comment · Reviewer_8g86 · 2025-11-25
> > **After rebuttal response**
> >
> > Thanks to the authors for providing their response.

---

### Official Review · Reviewer_o6TZ · 2025-11-07

**Soundness:** 3
**Presentation:** 1
**Contribution:** 2
**Rating:** 4
**Confidence:** 3

**Summary:**

This paper present a novel hierarchical reinforcement learning algorithm that generalize flat action-value Q to a new hierarchical W function, allowing the agent to execute actions according to an hierarchical strategy. The idea is inspired by the model of the brain called projective simulation, and each node in the hierarchy represent a separate task. The paper shows that this formulation side step the need to learn a termination function like we do in the option framework and is possible to learn all the tasks provided that the strategy could be represented in terms of a hierarchical clip structure. The learning algorithm for this new W function is extended to both the classic off-policy learning with an update rule derived from Q learning and to the actor-critic case by means of generalizing the policy gradient.

**Strengths:**

- The paper tackles the crucial and long-standing goal of developing effective hierarchical reinforcement learning (HRL) agents. This is a highly relevant research direction with the potential to significantly advance the field's ability to solve complex, long-horizon problems.

- While the theoretical development is dense, the simple and well-executed experiments provide an intuitive grounding for the W-learning concept. This section effectively clarifies the method's practical application and benefits.

- The W-learning formulation is novel and elegantly sidesteps significant limitations of prior HRL frameworks. Notably, it removes the need to explicitly learn termination functions and it seems to be able to converge to the optimal policy without the risk of hierarchical suboptimality.

**Weaknesses:**

- The paper is confusing the exact structure of a task/ clip structure is vague and hard to grasp. The paper spends much time on proposing an off-policy and actor-critic learning algorithms but fails to introduce and explain very well the new concepts introduced for a strategy and a clip unit making the paper hard to follow. I would suggest to first explain in detail the new idea and the hierarchical structure needed for this learning algorithm together with his properties and limitations before moving on to practical algorithms for learning that.

- The paper importantly is missing a broad discussion on the differences between the proposed new method and the methods already proposed in the literature like Option framework, Max Q, Feudal, Option Critic. Adding this discussion could help to clarify the differences of the proposed method compared to previous methods and highlight the advantages / disadvantages of the proposed methodology.

- The experiments are limited to a very easy discrete tabular domain, and the comparison is only against flat Q learning and only using the off-policy variant of W learning algorithm.

- One of the main challenges modern HRL is facing is to move from hand-crafted hierarchies to learned hierarchies, this paper yet introduce a new framework that still relies on hand-crafted hierarchies and a broader discussion on how this framework could be extended to learned hierarchies is needed.

**Questions:**

See Weaknesses points.

---

> ### Author Response · Authors · 2025-11-19
>
> Thanks for your valuable comments.
>
> • Yes, I agree the clip structure is confusing, and the goal of our work is to draw some useful ideas from it and make them concrete.     But instead of presenting the definition for the clip structure with strategy at the beginning, I choose to illustrate the rules for drawing the clip network at the beginning of the experiment (see page 6 around line 285), which fixes the gap between the theoretical construction and practical application.
>
> • Some discussions about connections between our work and other hierarchical learning methods are added in the new section (Related work at page 9 and 10).
>
> • Two more experiments about REINFORCE and REINFORCE with Baseline are added at page 9.
>
> • A broad discussion about strategy discovery is also added in the new section (page 9 and 10). Briefly speaking, one can start from a fully connected network and shape the architecture by cutting off legs with low transition probability during the pretraining procedure.

---

### Author Response · Authors · 2025-11-19
**Corrections for the new version**

Three main changes have been implemented in the new version

• Two more experiments based on the policy gradient method (REINFORCE and REINFORCE with baseline) are added at page 9. And we put the experiment results for double learning and n-step learning to appendix D.

• Section 4 about related works is added at page 9 and 10. The new section discusses differences between W-learning and other hierarchical learning methods and provides a broad way for strategy discovery. Also discusses further possible combination with deep learning.

• More explanations about the strategy and the clip structure are added at page 6, between line 285 and 290.

Minor typos are also corrected.

---

### Comment · Area_Chair_ihP4 · 2025-11-25

Dear Reviewers,

This is a gentle reminder to please take a moment to review the authors’ rebuttal for the manuscript currently under your evaluation. Your timely feedback will help us proceed with the next steps in the review process.

Thank you for your time and assistance.

Best regards,
AC

---

### Author Response · Authors · 2025-12-02
**Rebuttal Summary**

We would like to thank reviewers for their valuable comments. Here are some common issues raised by reviewers.
- Experiments are simple and only focus on the flat Q-learning and W-learning.
- Not enough clarification on the motivation or the setup of the hierarchical clip network.
- Lack of discussion on the relationship (advantages/disadvantages) with other hierarchical reinforcement learning methods.
- Lack of discussion on strategy discovery.
- There are other issues like the interpretation of parameters and optimality of policies.

To address those weaknesses
- We have added three more experiments including hierarchical REINFORCE, REINFORCE with Baseline and Actor-Critic. (see page 9)
- Additional clarification about the hierarchical clip network is added between the theoretical construction and experiment. (around line 285)
- A new section, Related Work, is added which discusses the connection between W-learning and other hierarchical learning methods, including a broad discussion on strategy discovery. Discussions on the optimality and deep learning are also added. (at page 9 and10)

---

### Meta-Review · Area_Chair_bW8x · 2026-01-05

**Summary:**

Most reviewers are concerning about the experiment results are not enough and fair. I agree with them and believe they are not fully addressed by the rebuttal. Hence I recommend for rejection.

**Reviewer Concerns:**

The reviewers have the following major concerns:

(1). Multiple reviewers (o6TZ, 6yDF) found the core concepts of "Clips" and "strategy" ($\omega$) to be vague, poorly defined, and hard to grasp. They noted that the paper failed to explain the precise hierarchical structure required.

(2). Reviewers criticized the loose connection to "brain models" (projective simulation) as empty statements without substance (GJFn, 6yDF).

(3). A major critique was the lack of discussion on established hierarchical RL (HRL) frameworks like Options, MAXQ, Feudal RL, or Option-Critic (o6TZ, 6yDF).

(4). Experiments were limited to very simple, tabular grid-world navigation tasks (o6TZ, 8g86, GJFn). The primary comparison was only against flat Q-learning. Reviewers requested comparisons against other HRL methods or stronger baselines (o6TZ, GJFn).

(5). Reviewer GJFn argued the method was "given" extra information (prior knowledge of the task/optimal policy without obstacles) that the baseline did not have, making the comparison unfair.


After the rebuttal, although some concerns are addressed, a few important ones are still outstanding:

The experiments remain limited to simple navigation tasks. There is no indication that the method was tested on complex, high-dimensional, or continuous control domains (like Mujoco/Atari), which reviewers implied were necessary for "modern" HRL evaluation.

The authors acknowledged they fed extra information to the agent but defended it by arguing that "teaching" agents strategies is a valid HRL goal. This philosophical defense may not satisfy reviewers looking for apples-to-apples algorithmic comparisons.

**Reviewer Scores:**

I believe most reviewers will maintain their scores.

---

### Decision · Program_Chairs · 2026-01-26

Reject